# Silicon Nanoparticles and Apoplastic Protein Interaction: A Hypothesized Mechanism for Modulating Plant Growth and Immunity

**DOI:** 10.3390/plants14111630

**Published:** 2025-05-27

**Authors:** Guopeng Miao, Juan Han, Taotao Han

**Affiliations:** 1Department of Bioengineering, Huainan Normal University, Huainan 232038, China; huanruyue@126.com (J.H.); hantt748@nenu.edu.cn (T.H.); 2Key Laboratory of Bioresource and Environmental Biotechnology, Anhui Higher Education Institutes, Huainan Normal University, Huainan 232038, China

**Keywords:** silicon nanoparticles, ROS, growth promotion, disease resistance, apoplastic proteins, protein coronas

## Abstract

Silicon nanoparticles (SiNPs) have emerged as multifunctional tools in sustainable agriculture, demonstrating significant efficacy in promoting crop growth and enhancing plant resilience against diverse biotic and abiotic stresses. Although their ability to strengthen antioxidant defense systems and activate systemic immune responses is well documented, the fundamental mechanisms driving these benefits remain unclear. This review synthesizes emerging evidence to propose an innovative paradigm: SiNPs remodel plant redox signaling networks and stress adaptation mechanisms by forming protein coronas through apoplastic protein adsorption. We hypothesize that extracellular SiNPs may elevate apoplastic reactive oxygen species (ROS) levels by adsorbing and inhibiting antioxidant enzymes, thereby enhancing intracellular redox buffering capacity and activating salicylic acid (SA)-dependent defense pathways. Conversely, smaller SiNPs infiltrating symplastic compartments risk oxidative damage due to direct suppression of cytoplasmic antioxidant systems. Additionally, SiNPs may indirectly influence heavy metal transporter activity through redox state regulation and broadly modulate plant physiological functions via transcription factor regulatory networks. Critical knowledge gaps persist regarding the dynamic composition of protein coronas under varying environmental conditions and their transgenerational impacts. By integrating existing mechanisms of SiNPs, this review provides insights and potential strategies for developing novel agrochemicals and stress-resistant crops.

## 1. Introduction

With the growth of the population, increasing food production has increasingly become an urgent issue facing humanity. As one of the promising strategies for plant protection and enhancing crop yields, nanotechnology has garnered widespread attention from global researchers in recent years [1,2]. Nanomaterials provide effective pest and pathogen control at substantially lower dosages—thus replacing more toxic chemicals and mitigating environmental pollution [3]. Nano-oxides such as ZnO, CuO, Fe_2_O_3_, and SiO_2_ have demonstrated numerous exceptional functionalities in agriculture, including enhancing seed germination [4], promoting plant growth [5,6,7], enhancing plant resistance to abiotic [8] and biotic stresses [9], and reducing heavy metal uptake in crops [10]. For instance, the foliar application of ZnO nanoparticles has been shown to promote tomato plant growth and yield while improving fruit quality and nutritional value [11], whereas Fe_2_O_3_ nanoparticles significantly enhance potato resistance to early blight pathogens [12].

Many metal oxides, such as silver (Ag), copper (Cu), and selenium (Se), can also be directly employed as bactericides and fungicides [13]. In addition, nano-oxides have been utilized as carriers for agrochemicals to enhance both pesticide efficacy and application efficiency [14] and as vectors for genetic material delivery to facilitate crop genetic modification [15]. Nanoclays and carbon nanotubes, for example, can improve soil properties by increasing water-holding capacity, adsorbing pollutants, and modulating microbial communities [16,17]. Non-metallic nano-oxides—such as nano-phosphorus and nano-sulfur—have similarly demonstrated activities that promote plant growth and confer resistance to environmental stresses [18,19,20].

Despite these promising applications, the use of nanoparticles in agriculture entails multiple risks. Studies have indicated that they may exert both physical and physiological toxicity on crops, resulting in diminished fruit yield and impaired plant growth. Moreover, farmers—who are the primary handlers of nano-enabled agrochemicals—face potential hazards from inhalation and dermal exposure. To date, the body of research on nanoparticle toxicity remains limited, raising concerns about unintended impacts on plants, soil microorganisms, animals, and even human health [21]. Silica degrades into a non-toxic compound called monosilicic acid, posing no ecological risks, making silicon nanoparticles (SiNPs) a relatively safe approach for enhancing crop yield and productivity in agricultural applications [22]. Therefore, compared to other nanoparticles, SiNPs not only exhibit the aforementioned characteristics but are also widely regarded as safer and more cost-effective to produce [23,24,25,26].

Despite the increasing number of studies confirming the potential of nanomaterials in improving crop performance, major knowledge gaps remain regarding the molecular mechanisms underlying their efficacy. Most nano-enabled agricultural products have reached this stage through empirical successes rather than a deep understanding of how they interact with plant physiology. Current challenges include identifying how nanoparticles are perceived and processed by plants, and knowledge gaps persist in particular around how nanoparticles trigger plant signaling pathways and stress responses at the molecular level [27]. Addressing these gaps is crucial for the rational design of next-generation agro-nanomaterials. This review, therefore, first provides a brief overview of the agricultural benefits of SiNPs and the existing mechanisms of action reported to date. Subsequently, based on observed phenomena, it analyzes potential underlying causes and proposes a new hypothesis that SiNPs may influence various plant physiological activities by interacting with apoplastic proteins (i.e., forming a “protein corona” in the plant’s extracellular space). Exploration of the fundamental mechanisms underlying nanomaterial efficacy may uncover novel targets, thereby facilitating the development of innovative chemical pesticides and the engineering of genetically modified crops, rather than merely optimizing formulations, application methods, and dosages.

## 2. Efficacy and Mechanisms of SiNPs in Improving Plant Growth and Stress Resistance

### 2.1. Promoting Plant Growth

Numerous pot and field experiments have confirmed that SiNPs can enhance germination rates, biomass, photosynthetic activity, and yields in various crops [28], such as tomatoes [29], wheat [30], potatoes [31], and rice [32]. Regarding the underlying mechanisms, studies suggest that SiNPs promote crop growth by balancing nutrient uptake [33], regulating carbon/nitrogen/phosphorus balance [34], and modulating auxin synthesis and carbohydrate metabolism [30].

For example, SiNPs have been shown to modify soil physicochemical properties, thereby significantly enhancing the uptake and translocation efficiencies of macronutrients such as phosphorus, potassium, and calcium in sorghum and other crops, which in turn promotes biomass accumulation and yield increases [35,36]. Additionally, SiNPs reduce leaf transpiration rates and improve root hydraulic conductivity, resulting in higher water-use efficiency and increased relative leaf water content in tomato and wheat [37,38]. Concurrently, SiNPs regulate endogenous indole-3-acetic acid (IAA) and gibberellin (GA) levels, activate cell-cycle-related genes, and stimulate cell elongation and division, thereby accelerating the proliferation and growth of both root and shoot tissues [39,40]. Moreover, the deposition of silicon–polysaccharide complexes within the middle lamella and cell wall matrix leads to the formation of rigid silica microstructures, markedly increasing cell wall mechanical strength and rigidity [41,42].

### 2.2. Enhancing Plant Tolerance to Abiotic Stress

Field crops are rarely able to achieve their genetically determined theoretical growth and yield potential due to fluctuating environmental conditions and varying degrees of stress [43]. A widely reported mechanism is that SiNPs regulate the activity of multiple antioxidant enzymes, including superoxide dismutase (SOD), catalase (CAT), and peroxidase (POX), reducing reactive oxygen species (ROS) generated under environmental stress and thereby improving crop growth [28,44]. This mechanism has been widely reported in crops such as spinach [45], wheat [46], rice [47], and maize [48] and in the mitigation of various abiotic stresses, including drought [49], heat [50], cold [51], and heavy metal toxicity [52]. However, the fundamental reasons behind SiNPs’ influence on antioxidant enzyme activity remain poorly understood [53].

As a distinct type of environmental stress, heavy metal accumulation not only restricts crop growth but also severely compromises agricultural product quality. Similar to their role in combating other environmental stresses, SiNPs mitigate the impact of heavy metals—such as lead [54], cadmium [46], aluminum [55], and chromium [56]—on plant growth by enhancing antioxidant defense mechanisms. Furthermore, SiNPs reduce heavy metal uptake and translocation through direct binding with heavy metals in the soil, deposition around endodermal cells to thicken the Casparian strip, and regulation of transporter protein expression and activity [57].

Since heavy metals are primarily transported systemically via transporter proteins in the symplastic pathway, the influence of SiNPs on these proteins has drawn significant attention. Plant cells possess several classes of heavy metal transporter proteins, including HMAs (heavy metal ATPases), Nramps (natural-resistance-associated macrophage proteins), CDFs (cation diffusion facilitators), ZIPs (ZRT, IRT-like proteins), and ABCs (ATP-binding cassette transporters) [58]. Numerous studies have demonstrated that SiNPs regulate heavy metal transport by modulating the transcription and activity of these transporter proteins, though the precise mechanisms remain unclear [57,59].

### 2.3. Enhancing Plant Resistance to Biotic Stress

SiNPs alleviate biotic stress in plants by reinforcing physical barriers, enhancing the synthesis of defensive compounds, activating salicylic acid (SA) and jasmonic acid (JA) signaling pathways, and modulating the expression of defense-related genes [28]. Long-term silicon application leads to its deposition beneath the cuticle, forming a secondary barrier outside the cell wall [60], and strengthens mechanical rigidity by binding to cell walls [61], thereby hindering pathogen invasion and spread. It is noteworthy that nano-silicon does not directly inhibit the growth of bacterial or fungal pathogens; rather, it confers such protection via indirect mechanisms [62]. At the physiological level, SiNPs interact with multiple signaling pathways, including those involving abscisic acid, gibberellins, auxins, JA, and SA [36]. Among these, increased SA synthesis and the induction of SA-mediated signaling pathways are the primary routes through which SiNPs enhance localized and systemic defense responses. For example, in *Arabidopsis*, SiNPs’ treatment triggers SA synthesis and upregulates the expression of SA pathway marker genes AtPR-1 and AtPR-5, mimicking the resistance response activated by avirulent pathogen *Pseudomonas syringae* [63]. A study by Du et al. [64] further demonstrated that foliar application of SiNPs stimulates SA-dependent immune mechanisms in rice, protecting it from *Magnaporthe oryzae* infection. Similar SA synthesis and accumulation phenomena have also been observed in wheat [65] and peanuts [66]. Moreover, SiNPs can enhance plant resistance to pathogens by modulating the composition and abundance of endophytic microbial communities [67].

SiNPs also induce the synthesis of secondary metabolites, critical components of plant defense against insect herbivory and pathogen attacks. Examples include diterpenoid phytoalexins like momilactones in rice [68], flavonoid-based antifungal compounds in cucumbers [69], and antimicrobial glycosides in wheat [70]. These effects are likely mediated by enhanced expression of downstream defense-related genes [71]. Additionally, SiNPs exhibit insecticidal activity through mechanisms such as ROS release and mechanical damage to insect cuticular cells [72].

In summary, SiNPs primarily promote plant growth by modulating hormonal signaling and altering nutrient uptake, mitigate abiotic stress through the suppression of ROS bursts, and enhance biotic stress resistance by upregulating SA biosynthesis and activating its downstream signaling pathways (Table 1). Despite the elucidation of these diverse mechanisms, the initial site of action by which SiNPs interact with plant physiological processes to trigger growth-promoting and stress-alleviating responses has yet to be identified.

## 3. The Size of SiNPs Profoundly Influences Distribution and Toxicity in Plants

To explore potential interaction sites, a thorough understanding of SiNP uptake and distribution in plants is essential. Nanoparticles can enter plants via stomata, root hairs, surface wounds, or seeds’ micropyles and intercellular pores, followed by transport through apoplastic and/or symplastic pathways [77,78]. Cellular entry of nanoparticles largely depends on their size: since plant cell wall pores average below 10 nm, particles with diameters of 3–5 nm have been reported to passively enter cells and vascular tissues via osmotic pressure or capillary action, while those larger than 10–20 nm are mostly restricted to the root endodermis or remain in leaf intercellular spaces (though some nanoparticles may enter cells through root tips, root hairs, or lateral root junctions lacking mature Casparian strips but fail to undergo upward translocation) [53]. This phenomenon has been documented in studies utilizing electron microscopy or energy-dispersive X-ray spectroscopy [63,64,79].

Notably, particle size not only affects accumulation sites and transport dynamics but also significantly influences the phytotoxicity of SiNPs [77]. Some studies report phytotoxic effects of SiNPs, depending on particle size, surface area, concentration, and plant species. From a toxicological perspective, particle size and surface area are critical factors. Smaller SiNPs with larger surface areas penetrate cellular compartments more rapidly, leading to higher toxicity compared to their bulk counterparts [80,81]. For instance, 10 nm silica particles significantly impair diatom growth and photosynthetic pigment synthesis more than larger particles [82], while monosilicic acid at equivalent concentrations exhibits greater toxicity to *Arabidopsis* than 50 nm SiNPs [63]. Intriguingly, toxicity responses of many nanoparticles are often linked to oxidative stress [81,83]. According to El-Shetehy et al. [63], high concentrations of Si(OH)_4_ induce oxidative stress in leaves, resulting in chlorophyll degradation, whereas SiNPs (50 nm) at concentrations as high as 1000 mg L^−1^ show no such phytotoxicity. Additionally, nanomaterials, including SiNPs, increase oxidative stress in mammalian cells [84]. However, as previously discussed, ROS reduction is a key mechanism by which SiNPs promote plant growth and enhance abiotic stress resistance.

## 4. SiNPs May Influence Apoplast Function via Protein Corona Formation

From the foregoing discussion, it is apparent that, owing to their limited entry into the symplastic pathway, larger SiNPs predominantly exert their effects by modulating apoplastic functions (Figure 1). The apoplast serves as a dynamic interface essential for plant growth and structural integrity. It facilitates cell wall synthesis and remodeling through enzymes like expansins (promoting wall loosening) and POX (mediating lignin cross-linking) [85]. Key molecules, including hormones (e.g., auxins, cytokinins) and ROS, regulate cell elongation, division, and differentiation [86]. The pH and redox status of the apoplast further modulate developmental signals, such as stomatal aperture and root growth [86]. The apoplast is also the frontline battlefield against biotic and abiotic stresses [87]. Antimicrobial proteins [88] and ROS [89] directly inhibit pathogens, while antioxidants detoxify ROS to prevent oxidative damage [90]. During abiotic stress, apoplastic antioxidative enzymes and osmolytes (e.g., proline, phenolic compounds) maintain cellular homeostasis. Hormones like ABA and SA coordinate systemic defenses, while apoplastic pH shifts and ion fluxes regulate stress signaling [90]. As key effective molecules in plant immune defense systems, PR proteins combat pathogen invasion through enzymatic catalysis, membrane disruption, and signal transduction. They directly degrade pathogen cells, promote cell wall lignification to strengthen physical barriers, and participate in programmed cell death to limit infection spread [91]. Additionally, the apoplast harbors diverse signaling proteins. For instance, apoplastic PLCPs (papain-like cysteine proteases) generate Zip1 (*Zea mays* immune signaling peptide 1), a polypeptide that activates SA-dependent defense responses by cleaving propeptide precursors [92]. In *Arabidopsis*, the natriuretic peptide-like protein AtPNP-A suppresses SA synthesis and antagonizes SA signaling upon binding to its plasma membrane receptor, thereby weakening resistance to pathogens [93].

When nanomaterials interact with biological environments, they adsorb a layer of proteins on their surfaces via non-covalent interactions (e.g., electrostatic forces, hydrophobicity, hydrogen bonding, and π-π stacking), forming a structure referred to as a protein corona. Protein adsorption can induce conformational changes, exposing or masking functional epitopes. For example, hydrophobic surfaces may promote protein unfolding, thereby activating or inhibiting biological functions [94]. While the protein composition of the corona depends on nanoparticle physicochemical properties and environmental conditions (e.g., pH, ionic strength, temperature), specific nanomaterials can drive selective adsorption through surface characteristics (e.g., chirality, functional ligands), forming a “personalized” protein corona with distinct specificity [95].

Based on summarized mechanisms in Section 2, through interaction with apoplastic proteins, SiNPs may elevate cellular redox levels, enhance SA synthesis, and regulate transporter protein expression and activity. For example, the adsorption of PR proteins by SiNPs may compromise their pathogen defense efficacy; similarly, the binding of antioxidant enzymes could perturb apoplastic ROS homeostasis, while the interaction of SiNPs with PLCPs, Zip1, or PNP-A might directly disrupt their regulatory functions in SA biosynthesis.

## 5. SiNPs May Exert Multifunctional Effects via Modulating Apoplastic ROS Homeostasis

Apoplastic ROS and antioxidant enzyme systems play pivotal roles in critical physiological processes, including cell wall biosynthesis, growth regulation, antimicrobial defense, and activation of stress-responsive pathways. ROS in the plant apoplast principally comprise superoxide anions (O_2_^•−^), H_2_O_2_, hydroxyl radicals (•OH), singlet oxygen (^1^O_2_), and ozone (O_3_). Among these, O_2_^•−^ and •OH exhibit extremely high reactivity and very short half-lives, confining their oxidative damage or rapid scavenging to the site of generation. Singlet oxygen and ozone, owing to their chemical instability or limited modes of production, have not yet been systematically characterized for signaling roles [86]. In contrast, the relative stability and longer half-life of H_2_O_2_ permit its diffusion across cell walls and membranes, entry into the cytosol via aquaporins, and subsequent activation of Ca^2+^ channels, MAPK cascades, and SA-mediated defense and growth responses [96]. Crucially, the spatiotemporal regulation of H_2_O_2_ generation and removal—mediated by ascorbate peroxidase (APX), CAT, and glutathione peroxidase (GPX)—establishes redox signaling gradients that enable localized signals to be amplified systemically, a cascade function that more reactive ROS cannot sustain [97,98].

Emerging evidence highlights the centrality of ROS in SiNP-mediated stress mitigation mechanisms. A recent study demonstrated that RBOH (respiratory burst oxidase homolog)-dependent ROS generation is indispensable for SiNP-induced lead detoxification: chemical inhibition of ROS production or RBOH signaling pathways severely compromised SiNP’s ability to enhance antioxidant enzyme activity and counteract Pb toxicity [99]. Our preliminary investigations revealed that SiNPs significantly inhibit the enzymatic activities of apoplastic SOD and POX in rice through surface adsorption (unpublished data). Moreover, a review of the above-mentioned studies on the oxidative stress toxicity induced by SiNPs—demonstrating that smaller-sized nanoparticles more readily elicit ROS stress responses—further confirms the stimulatory effect of SiNPs on ROS production. These findings provide mechanistic evidence that SiNPs elevate apoplastic ROS levels by suppressing antioxidant enzyme functionality. As versatile signaling molecules, apoplastic ROS concentrations exhibit dynamic correlations with plant growth modulation, developmental programming, and stress resilience.

### 5.1. Apoplastic H_2_O_2_ Promotes Plant Growth and Enhances Tolerance to Abiotic Stresses by Modulating Intracellular Redox Homeostasis

Apoplastic H_2_O_2_ homeostasis is achieved through a dynamic balance of enzymatic production—primarily by plasma membrane RBOHs whose activity is fine-tuned by Ca^2+^-dependent phosphorylation and salicylic acid signaling—and scavenging by class III POX, with peroxiporin aquaporins facilitating bidirectional H_2_O_2_ diffusion across the plasma membrane to regulate signal amplitude and duration [100]. Elevated apoplastic H_2_O_2_ levels trigger compensatory activation of intracellular antioxidant systems, thereby influencing seed germination, plant growth, and resistance to abiotic stresses.

In numerous plant species, apoplastic H_2_O_2_ has been demonstrated to serve as a pivotal signal for breaking seed dormancy and promoting germination. In *Jatropha curcas*, seed priming with H_2_O_2_ stimulates L-cysteine desulfhydrase activity, inducing hydrogen sulfide (H_2_S) production; the two reactive species act synergistically to amplify germination signals and increase germination rates [101] In *Zinnia elegans*, exogenous H_2_O_2_ promotes germination in a dose-dependent manner by oxidizing inhibitory compounds within the pericarp, thereby relieving both mechanical and chemical constraints on radicle protrusion [102]. During the early germination of *Pisum sativum* seeds, a transient apoplastic H_2_O_2_ peak—mediated by cell wall peroxidases—diffuses into adjacent cells to activate expansins and aquaporins, facilitating testa rupture and water uptake; this underscores the role of apoplastic H_2_O_2_ as a diffusible signaling molecule [103].

Beyond germination, apoplastic H_2_O_2_ also contributes to vegetative growth. For instance, maize seedlings pretreated with micromolar concentrations of H_2_O_2_ exhibited up to a 15% increase in dry matter accumulation and enhanced distribution of mineral nutrients, effects that were attributed to improved osmotic balance and membrane stability [104]. Similarly, soil amendment with H_2_O_2_ in paddy fields enhanced rice shoot and root biomass, likely by priming stress-responsive pathways and facilitating nutrient uptake [105]. Mechanistically, H_2_O_2_ has been demonstrated to induce the expression of cell cycle, redox regulation, and cell wall organization genes in *Arabidopsis* seedlings, accelerating cell proliferation and tissue growth under stress conditions [106]. In the research of Considine et al. [107], ROS elevation initiates signaling cascades that activate the TOR pathway, driving phosphorylation of downstream targets to promote cell proliferation, developmental plasticity, and environmental adaptation.

Apoplastic H_2_O_2_ plays a critical role in augmenting the overall redox potential of plants and bolstering their tolerance to abiotic stresses. For instance, exogenous H_2_O_2_ application in wheat seedlings enhances SOD, POD, CAT, and POX activities while increasing glutathione (GSH) and carotenoid concentrations, thereby improving salt stress tolerance [108]. Genetic studies in melon reveal that the knockout of CmPIP2;3, a plasma membrane H_2_O_2_ transporter, severely attenuates trehalose-mediated cold stress resistance by disrupting intracellular redox regulation [109]. The mechanism by which apoplastic H_2_O_2_ influences intracellular redox status is complex; it can indirectly increase cellular glutathione levels and redox capacity (GSH/GSSG ratio) by stimulating SA biosynthesis, thereby sustaining NPR1 (Non-expressor of Pathogenesis-Related genes 1)-dependent SA signaling [110,111]. Post-stress ROS dynamics typically exhibit a transient spike followed by a decline, with subsequent secondary ROS accumulation in plastids at reduced magnitudes [112]. Notably, effector-triggered immunity (ETI) induces explosive ROS bursts that synergize with SA through positive feedback loops, culminating in hypersensitive response (HR) and programmed cell death [111,113]. Based on this framework, we hypothesize that larger SiNPs may indirectly increase redox potential via apoplastic ROS signaling.

### 5.2. Apoplastic H_2_O_2_ Augments Biotic Stress Resistance

Apoplastic H_2_O_2_ fulfills multiple roles in plant defense against biotic stress: it acts both as a signaling molecule to trigger systemic resistance and as a direct antimicrobial agent. Locally accumulated H_2_O_2_ can oxidatively damage pathogen proteins and cellular structures, effectively inhibiting fungal colonization by species such as *Penicillium expansum* [114,115]. Concurrently, reactive oxygen bursts mediated by RBOHs and apoplastic oxidoreductases are well documented to enhance both local and systemic SA biosynthesis [111]. A recent breakthrough study demonstrated that H_2_O_2_ induces oxidative modification of the transcription factor CHE (CCA1 Hiking Expedition), thereby upregulating SA biosynthetic gene expression and activating SA-dependent signaling pathways, which amplifies plant immune responses [116]. Moreover, under severe pathogen challenge, apoplastic H_2_O_2_ can activate mitogen-activated protein kinase (MAPK) cascades to initiate programmed cell death and HR, rapidly forming a barrier that confines pathogen spread [117].

Apoplastic ROS not only translocate into cells via channel proteins but also interact with plasma membrane receptors. Wu et al. [118] demonstrated that extracellular H_2_O_2_ oxidizes extracellular cysteine residues of HPCA1 (hydrogen-peroxide-induced Ca^2+^ increases 1), triggering autophosphorylation and Ca^2+^ channel activation to regulate stomatal closure and systemic H_2_O_2_ signaling.

Whether in response to abiotic or biotic stress, apoplastic H_2_O_2_ induces systemic priming effects through multidimensional reprogramming at physiological, transcriptional, post-translational, metabolic, and epigenetic levels, enabling accelerated stress responses upon subsequent challenges [43]. Notably, SiNPs and other nanomaterials exhibit analogous priming capacities [119,120].

### 5.3. Redox Status Modulates Transporter Protein Functionality

Following adsorption and modulation of apoplastic protein activities, SiNPs may indirectly regulate transporter gene transcription via transcription factors or modulate transporter phosphorylation through alterations in kinase signaling pathways. Given that the impact of SiNPs on ROS is most extensively documented, we herein focus on how SiNP-mediated upregulation of apoplastic ROS levels influences transporter function.

RBOH-generated ROS regulates transporter activity at transcriptional and post-translational levels, thereby influencing heavy metal uptake and translocation [121,122]. Consistently, the transcriptional regulation and post-translational activity of numerous transporters are profoundly influenced by intracellular redox status [123]. HMA transporters play pivotal roles in Zn/Cd homeostasis: knockout of two homologous HMAs in tobacco reduces Cd translocation by >90% [124], while overexpression of OsHMA3 in rice sequesters Cd into vacuoles, enhancing Cd tolerance [125]. The C-terminal domains of HMAs harbor multiple cysteine residue pairs that likely serve as redox-sensitive regulatory sites [126,127].

ABC transporters, such as AtPDR8/12 (mediating Cd/Pb uptake in *Arabidopsis*) and OsPDR9/20 (Cd efflux in rice), contain phosphorylation motifs critical for functional modulation [128]. Our prior work demonstrated that ROS rapidly signals to PDR transporters via AGC kinases, altering their capacity to mobilize secondary metabolites in *Arabidopsis* [129]. Additionally, the transcriptional and post-translational regulation of Nramp, ZIP, and CDF families is redox-responsive [123]. Collectively, these findings suggest that SiNPs may indirectly modulate transporter activity by altering cellular redox status, ultimately affecting heavy metal dynamics. Supporting this hypothesis, Karimi-Baram et al. [99] reported that RBOH/ROS inhibition abolished SiNP-mediated Pb accumulation blockade.

## 6. Hypothesized Action Mechanisms of SiNPs

The current understanding of SiNP efficacy remains constrained by unresolved mechanistic questions:Redox Modulation Paradox: How do SiNPs enhance cellular redox potential, and why do smaller particles with higher concentrations preferentially induce oxidative damage?SA Signaling Activation: Through what molecular routes do SiNPs stimulate SA biosynthesis and subsequent signaling cascades?Transporter Regulation: By what means do SiNPs modify heavy metal transporter activity (e.g., HMA and ABC transporter families) to restrict heavy metal uptake and translocation?

Synthesizing preceding discussions, we propose the model shown in Figure 2. Firstly, both SiNPs and silicic acid trigger ROS generation. Larger SiNPs predominantly elevate extracellular ROS due to restricted cellular entry, which feedback-regulates intracellular antioxidant systems to enhance redox buffering capacity. Smaller particles penetrate cells more readily, overwhelming antioxidant defenses and causing oxidative stress. Apoplastic ROS elevation may indirectly stimulate SA biosynthesis, while direct SiNP-protein interactions (e.g., with signaling components like PNP-A or PLCP) could activate or inhibit SA regulatory nodes. Specifically, SiNP binding to PNP-A may relieve its SA-suppressive effects by blocking its interaction with membrane receptors. For the PLCP-Zip1 axis, SiNP interaction might exhibit dual effects: conformational changes in PLCP induced by SiNP adsorption could either enhance or suppress PLCP proteolytic activity, thereby differentially regulating Zip1 production and downstream SA signaling. Finally, redox-sensitive transporter families (HMA, ABC, etc.) and transcription factors may be indirectly modulated by SiNP-induced shifts in cellular redox equilibrium.

Notably, the adsorption of apoplastic proteins by SiNPs may elicit multifaceted physiological responses. The proposed model, however, may not fully encapsulate all mechanistic possibilities, particularly given the dynamic reciprocity between nanoparticle surface chemistry and plant stress-adaptive plasticity. Consequently, these hypotheses require rigorous experimental validation. We highlight protein corona profiling as a central validation strategy, given its unique advantages in probing plant–nanomaterial interactions.

Protein corona profiling entails isolating SiNPs from plant tissues after exposure and identifying the proteins adsorbed on their surfaces (the “corona”) via proteomic analyses. This approach directly reveals the molecular interface between SiNPs and plant cells, pinpointing which apoplastic proteins physically associate with the nanoparticles. Such information is invaluable because these adsorbed proteins can fundamentally alter the nanoparticles’ biological identity, influencing how the SiNPs are recognized and transported within plant tissues. In animal and environmental systems, extensive research has shown that the corona composition dictates nanoparticle fate and effects, but in plant systems, this phenomenon remains comparatively under-characterized [130]. By applying protein corona profiling in plants, we can capitalize on this concept to uncover which specific proteins mediate SiNP-induced responses. This method offers an unbiased, system-level view of nanoparticle-protein interactions, in contrast to traditional physiological assays or gene expression studies that might indicate downstream effects but cannot directly identify the initial molecular targets of SiNPs. In essence, protein corona profiling helps answer “Which proteins does the nanoparticle bind or affect first?”—a critical unknown in elucidating SiNP mechanisms.

To clarify the mechanisms of SiNPs, alternative methods exist, but they address different facets of the problem. For example, redox flux mapping (using ROS-sensitive fluorescent probes or biosensors) can confirm that SiNPs alter ROS dynamics in space and time, and structural–functional studies (e.g., X-ray crystallography or spectroscopy on SiNP-protein complexes) can reveal how nanoparticle binding changes the conformation or activity of a particular protein. By directly focusing on the nanoparticle-bound proteome, protein corona profiling provides a mechanistic shortcut to link SiNP treatment with specific molecular actors. The insights gained are highly actionable: for instance, if profiling reveals that a certain antioxidant enzyme or signaling peptide consistently adsorbs to SiNPs, researchers can then investigate that protein’s functional role (e.g., does SiNP binding inhibit its activity and thereby trigger a signaling cascade?).

Deciphering the molecular choreography of SiNP-plant interactions will advance the rational design of nano-enabled agrochemicals and inform breeding strategies for stress-resilient crops through novel molecular targets.

## Figures and Tables

**Figure 1 plants-14-01630-f001:**
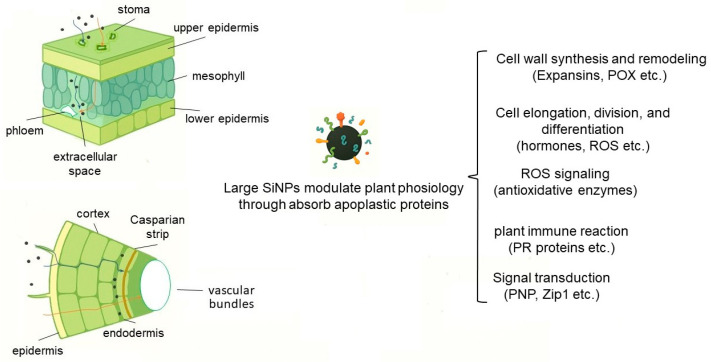
Schematic representation of the apoplastic localization and mode of action of large SiNPs in leaf and root tissues. Following foliar or rhizospheric application, SiNPs are largely retained in the extracellular (apoplastic) spacebetween the upper and lower epidermis in leaves and outside the endodermis in rootsdue to restricted symplastic entry. Here, they adsorb and modulate apoplastic proteins to regulate diverse physiological processes, including cell wall synthesis and remodeling (e.g., expansins, POX), cell elongation and division (via hormonal and ROS signaling), antioxidant defense (antioxidative enzymes), pathogen-triggered immune responses (PR proteins), and signal transduction pathways (e.g., PNP, Zip1).

**Figure 2 plants-14-01630-f002:**
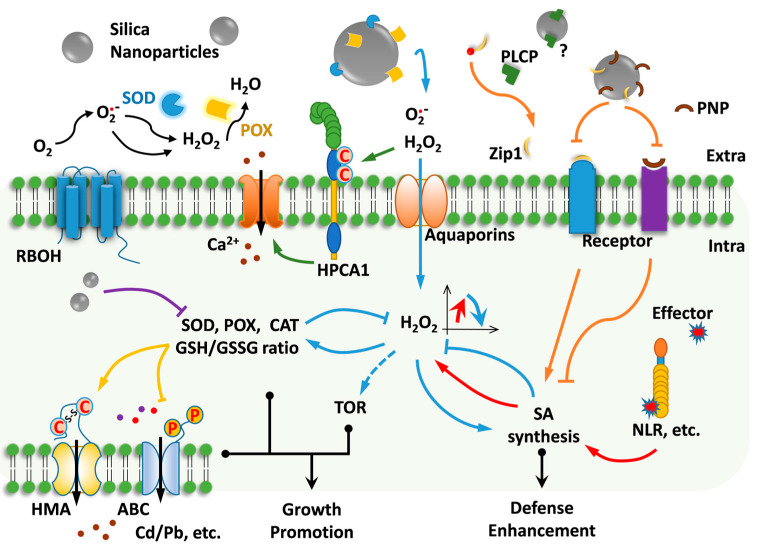
Schematic diagram of the scientific hypothesis mechanism. **Blue Arrow:** The activity of intercellular antioxidant enzymes is inhibited after binding with nano-silicon, leading to the accumulation of intercellular ROS. The accumulated ROS are transported into the cell via channel proteins, thereby enhancing intracellular reducing capacity and promoting SA synthesis. **Orange Arrow:** Intercellular signaling proteins PNP and Zip1 are adsorbed by nano-silicon, preventing their binding to receptors. This results in the activation or inhibition of SA synthesis, respectively. The adsorption of PLCP by nano-silicon may either promote or inhibit its protease activity. **Red Arrow:** Effector molecules acting on intracellular receptors (e.g., NLRs, Nucleotide-binding Leucine-rich Repeat receptors) may induce SA synthesis, potentially forming a positive feedback loop between ROS and SA. **Yellow Arrow:** Increased intracellular reducing capacity may alter the conformation of transport proteins (e.g., HMA and ABC) through post-translational modifications, thereby affecting heavy metal uptake, storage, and translocation (e.g., enhanced vacuolar transport). **Purple Arrow:** Small-sized nano-silicon particles entering the cell may directly inhibit intracellular antioxidant enzyme activity. **Green Arrow:** The HPCA1 receptor is activated upon oxidation by extracellular H_2_O_2_, further triggering Ca^2+^ ion channels to propagate systemic H_2_O_2_ signaling. **Notes:** Although slower than the catalytic reaction of SOD enzymes, O_2_^•−^ can spontaneously convert to H_2_O_2_. Elevated extracellular ROS levels may induce a transient increase followed by a decrease in intracellular ROS content.

**Table 1 plants-14-01630-t001:** Typical examples illustrating the efficacy and underlying mechanisms of SiNPs.

SiNP Size	Application Method	Biological Effect	Proposed Mechanism	Ref.
~50 nm	Foliar spray	Enhanced systemic acquired resistance in *A. thaliana*	Nanoparticles enter through stomata into the apoplast and activate SA signaling, upregulating PR-1 and PR-5 expression.	[63]
~200 nm	Foliar spray	~27.7% reduction in lesion size of *Fusarium* head blight on wheat ears	Formation of a physical barrier on the leaf surface; increased POD and SOD activities, reduced CAT and DHAR, lower ROS accumulation, and upregulation of PR genes and SA levels.	[65]
~30 nm	Root drench	33.3% yield increase in rice under salt stress; higher chlorophyll and root growth	Improved water and nutrient uptake by roots; upregulation of SOD, POD, and CAT activities and lowered MDA content to alleviate oxidative damage under salinity.	[73]
~20 nm	Soil drench	Improved growth and biomass of bamboo under lead stress	SiNPs enhance capacity of SOD, POD, CAT, and glutathione reductase and reduce heavy metal accumulation.	[54]
~10–50 nm	Foliar spray	Increased fresh and dry weight and chlorophyll levels in wheat under salt stress	Promotion of proline and free amino acid synthesis; enhanced nutrient accumulation; upregulation of SOD, CAT, and POD activities, leading to reduced oxidative damage.	[74]
10–20 nm	Root drench	Recovery of growth, photosynthetic efficiency, and biomass in maize under aluminum toxicity	Reduced activities of photorespiratory enzymes and NADPH oxidase, maintenance of redox balance; promotion of aluminum chelation and detoxification.	[48]
~10–17 nm and 110–120 nm	Seed soaking	Mean germination time reduced from 5.24 ± 0.29 d to 4.64 ± 0.29 d; seedling vigor (length and weight) improved	SiNPs enhance water imbibition by seeds and alter the external microenvironment.	[75]
~20–30 nm	Root drench	Increased biomass of spinach under lead pollution	Synergistic action with lead-tolerant bacteria; enhanced SOD, POD, and CAT activities; reduced MDA; decreased lead uptake and translocation from root to shoot.	[45]
40–60 nm	Foliar spray	~70% reduction in rice blast severity (*M. oryzae*) in rice	Elevated apoplastic SA levels; strong upregulation of PR genes; formation of a nanoparticle barrier around stomata that impedes pathogen entry.	[64]
~10–25 nm	Root drench	Increased biomass and reduced Cd content in wheat	Enhanced antioxidant defenses and induction of transporter gene expression to inhibit Cd translocation.	[76]
~20 nm	Foliar spray	Improved cold tolerance in tomato under chilling stress	SiNPs ameliorated the osmotic adjustment and antioxidant capacity of the plants.	[51]

## Data Availability

All data are contained within the article.

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
