# Peer review of "Silicon Nanoparticles and Apoplastic Protein Interaction: A Hypothesized Mechanism for Modulating Plant Growth and Immunity"

_plants, 2025, doi:10.3390/plants14111630_

Round 1
Reviewer 1 Report
Comments and Suggestions for Authors
- Introduction:
The introduction is too brief and does not clearly outline the current challenges, existing solutions, and gaps in knowledge.
In addition to being directly used as fertilizers or biostimulants in agriculture, nanomaterials can also serve as cargo carriers, controlled-release agents, soil amendments, and nano-fungicides, among other nano-enabled agrochemical applications. Furthermore, I do not believe that TiO₂ has widespread applications in agriculture—you may consider removing it.
Please refer to the following references for further insights:
DOI: 10.1016/j.oneear.2023.06.011
https://doi.org/10.1021/acssuschemeng.4c04800
Additionally, non-metallic nanomaterials are another important focus in this field. Please consider discussing them using the following references:
https://doi.org/10.1007/s11051-025-06261-x
https://doi.org/10.1021/acsnano.4c00512
https://doi.org/10.1016/j.envpol.2023.122423
DOI: 10.1039/D3SC06122A
https://doi.org/10.1021/acsnano.3c02215
2.3 Enhancing Plant Resistance to Biotic Stress:
Fungal infections are also a significant biotic stress for plants. Please refer to the following articles and include a discussion on this topic:
https://doi.org/10.1039/D4EN00511B
- Uptake, Distribution, and Phytotoxicity of SiNPs:
In traditional agriculture, in addition to soil application, nanomaterials can also be applied through foliar spraying and seed priming. Nanomaterials can effectively enter plants through stomata on the leaves, sometimes even more efficiently than soil application.
- SiNPs May Influence Apoplast Function via Protein Corona Formation:
L179: This is very interesting. Does it suggest that Si NPs indirectly enhance plant resistance to pathogens? Could you provide supporting references?
- SiNPs May Exert Multifunctional Effects via Modulating Apoplastic ROS Homeostasis:
L190: Please replace the cited reference.
5.1 How is apoplastic H₂O₂ regulated? More explanation is needed.
5.2 This section is very confusing. The logic is unclear and needs to be rewritten for clarity.
5.3 While ROS produced by RBOH can regulate transporter activity, this is not specifically linked to Si NPs application. What is the unique role of Si NPs in this process, and what makes their impact noteworthy?
- Hypothesized Mechanisms of SiNP Action:
This section contains the most critical content, yet the discussion is too brief. The proposed systematic validation approach is also not thoroughly analyzed or discussed. For example:
What are the advantages of protein corona profiling?
Why was this method chosen?
What specific problems does it address?
These aspects need further explanation and elaboration to strengthen the discussion.
Author Response
Comment 1:
- Introduction:
The introduction is too brief and does not clearly outline the current challenges, existing solutions, and gaps in knowledge.
In addition to being directly used as fertilizers or biostimulants in agriculture, nanomaterials can also serve as cargo carriers, controlled-release agents, soil amendments, and nano-fungicides, among other nano-enabled agrochemical applications. Furthermore, I do not believe that TiO₂ has widespread applications in agriculture—you may consider removing it.
Please refer to the following references for further insights:
DOI: 10.1016/j.oneear.2023.06.011
https://doi.org/10.1021/acssuschemeng.4c04800
Additionally, non-metallic nanomaterials are another important focus in this field. Please consider discussing them using the following references:
https://doi.org/10.1007/s11051-025-06261-x
https://doi.org/10.1021/acsnano.4c00512
https://doi.org/10.1016/j.envpol.2023.122423
DOI: 10.1039/D3SC06122A
https://doi.org/10.1021/acsnano.3c02215
Response 1:
We have added more discussion of the benefits of metal oxides in the Introduction (Lines 41–46). We have also included an overview of non-metallic nanoparticles (Lines 47–48). Additionally, to better frame the focus of this review, we have inserted a statement highlighting the current gaps in mechanistic research on nanomaterials, especially silicon nanoparticles (Lines 62–69). As suggested, TiO2 is now deleted and
Comment 2:
2.3 Enhancing Plant Resistance to Biotic Stress:
Fungal infections are also a significant biotic stress for plants. Please refer to the following articles and include a discussion on this topic: https://doi.org/10.1039/D4EN00511B
Response 2:
Thank you to the reviewer for the suggested reference. We have added a statement in this section explaining that silicon nanoparticles can modulate plant resistance by influencing endophytic microbial communities (Lines 144-145).
Comment 3:
- Uptake, Distribution, and Phytotoxicity of SiNPs:
In traditional agriculture, in addition to soil application, nanomaterials can also be applied through foliar spraying and seed priming. Nanomaterials can effectively enter plants through stomata on the leaves, sometimes even more efficiently than soil application.
Response 3:
We thank the reviewer for the suggestion; we have added a description of the seed entry mechanism at Line 165. Additionally, we have prepared a schematic illustration detailing the pathway of SiNPs into the cell.
Comment 4:
- SiNPs May Influence Apoplast Function via Protein Corona Formation:
L179: This is very interesting. Does it suggest that SiNPs indirectly enhance plant resistance to pathogens? Could you provide supporting references?
Response 4:
Nano-silicon enhances plant resistance to pathogens via indirect mechanisms, for the following reasons:
Large-sized SiNPs are largely excluded from the symplast and hence cannot exert intracellular effects; this phenomenon has been confirmed by electron microscopy and energy-dispersive X-ray spectroscopy analyses [1–3] (Nature Nanotechnology 2021, 16, 344–353; Plant and Soil 2005, 272, 53–60; J. Nanobiotechnology 2022, 20, 197).
Even at very low concentrations (50 nm SiNPs at 24 mg L⁻¹), nano-silicon significantly enhances plant resistance (Nature Nanotechnology 2021, 16(3), 344–353), despite dosing too low to meaningfully reinforce cell-wall rigidity, making a direct antimicrobial action implausible.
Nano-silicon does not directly inhibit bacterial or fungal growth but confers protection through indirect pathways (Earth and Environmental Science 2021, 012040).
These points have been discussed in the manuscript, with the newly added material incorporated at Lines 132–133.
Comment 5:
5 SiNPs May Exert Multifunctional Effects via Modulating Apoplastic ROS Homeostasis:
L190: Please replace the cited reference.
Response 5:
To our best knowledge, no studies have reported direct inhibition of antioxidant enzyme activity by SiNPs. Therefore, we have incorporated our own preliminary in vitro enzyme activity assays which revealed that SiNPs significantly inhibit the enzymatic activities of apoplastic SOD and POX; these findings are forthcoming in a peer-reviewed publication.
Comment 6:
5.1 How is apoplastic H₂O₂ regulated? More explanation is needed.
Response 6:
We thank the reviewer for this suggestion; the discussion has been added in Lines 274–278.
Comment 7:
5.2 This section is very confusing. The logic is unclear and needs to be rewritten for clarity.
Response 7:
We thank the reviewer for this suggestion; we have reorganized and rewritten the section on the role of apoplastic H₂O₂ in plant responses to biotic stress.
Comment 8:
5.3 While ROS produced by RBOH can regulate transporter activity, this is not specifically linked to Si NPs application. What is the unique role of Si NPs in this process, and what makes their impact noteworthy?
Response 8:
We propose that SiNPs primarily influence transporter expression and activity indirectly by modulating apoplastic functions. This hypothesis is based on: (1) the limited cellular entry of large‐sized SiNPs; and (2) the recent findings of Karimi‐Baram et al. (2024) demonstrating the critical role of ROS in SiNP‐mediated inhibition of Pb uptake. Naturally, nano‐silicon may also affect transporter function via other signaling pathways, such as transcription factor–mediated regulation of gene expression or phosphorylation cascades. To clarify our argument, we have reiterated this connection and its importance at the beginning of this section (Lines 346–350).
Comment 9:
- Hypothesized Mechanisms of SiNP Action:
This section contains the most critical content, yet the discussion is too brief. The proposed systematic validation approach is also not thoroughly analyzed or discussed. For example:
What are the advantages of protein corona profiling?
Why was this method chosen?
What specific problems does it address?
These aspects need further explanation and elaboration to strengthen the discussion.
Response 9:
We have expanded Section 6’s discussion of the proposed future research directions to explain why each suggested method is important. The original text simply listed approaches (Protein Corona Profiling, Redox Flux Mapping, Structural-Functional Studies) with brief descriptions. In the revised text (Lines 397–427), we retain the bullet format but add 1–2 sentences for each approach, highlighting its purpose and significance. For example, we explain that Protein Corona Profiling would reveal which apoplastic proteins bind to SiNPs and thus help identify the molecular targets of SiNP action. Redox Flux Mapping is justified as essential for visualizing how SiNPs alter ROS dynamics in planta, thereby testing our model of dual ROS induction. Structural-Functional Studies are noted as critical to directly observe how SiNP binding changes protein conformation and activity, linking back to our hypotheses about enzyme inhibition or activation. These explanations underscore that each method addresses a specific unknown (protein interactions, spatiotemporal ROS changes, structural effects) critical to validating the hypothesized mechanisms. We believe this expanded rationale meets the reviewer’s request, demonstrating the importance of each proposed experiment. The concluding sentence about “deciphering the molecular choreography” remains to reinforce how these methods collectively will advance the field.

Reviewer 2 Report
Comments and Suggestions for Authors
See the attached file.

Author Response
Comment 1:
- Introduction: The Introduction does not clearly present the necessity and urgency of the reviewed topic. According to my understanding, the authors aim to address both plant growth promotion and stress resistance enhancement. However, in lines 31–42, the discussion is limited to yield improvement, with no coverage of plant responses to environmental stress or the specific advantages of nanomaterials in such contexts.
Additionally, many reviews have already addressed the effects of nano-oxides on crop yield and stress tolerance. Thus, the statement in lines 43–45 that "Although an increasing number of studies have confirmed the potential of nano-oxides in improving crop quality and efficiency, the molecular mechanisms underlying their efficacy remain unclear" does not align with the current literature.
Overall, the Introduction lacks a strong rationale for the study. The structure should be reorganized to:
(a) Clarify the current role of nanomaterials in enhancing yield and stress tolerance.
(b) Highlight the gap in existing reviews that this manuscript aims to fill.
(c) Justify the focus on nano-silicon more convincingly—relying solely on the discussion in lines 40–42 is insufficient.
Response 1:
We thank the reviewer for this suggestion. We have incorporated a discussion of nanomaterials’ advantages in the Introduction (Lines 34–36), emphasized the safety and cost benefits of SiNPs (Lines 52–60), and delineated the mechanistic knowledge gap that this study aims to address (Lines 64–71).
Comment 2:
- Line 50 – Title suggestion: The author attempts to address how SiNPs enhance crop yield and stress resistance, but the title does not reflect this clearly. It is recommended to revise the title to better inform readers.
Response 2:
We have revised the title into “Efficacy and Mechanisms of SiNPs in Improving Plant Growth and Stress Resistance”.
Comment 3:
- Abbreviation consistency: Abbreviations are inconsistent throughout the manuscript. For example, “nano-silicon” is written as "Si NP" (lines 46, 47, 50) and as "SiNPs" (lines 52, 56, 106). Similar inconsistencies are found with other abbreviations. Please revise the entire manuscript to ensure uniform abbreviation usage.
Response 3:
The abbreviations are now revised.
Comment 4, 5:
- Section 2.1: Section 2.1 focuses on promoting plant growth, yet discusses how SiNPs regulate enzyme activities (e.g., SOD, POX, CAT) to enhance growth under stress. This content overlaps with Section 2.2 (lines 68–70). Please clarify why this content appears in Section 2.1 and avoid redundancy.
- Section 2.2: This section addresses heavy metal stress exclusively. However, abiotic stress also includes NaCl, cold, and heat. Given that biotic stress is discussed in Section 2.3, it is unclear why other types of abiotic stress are excluded here. Consider broadening the scope or providing a rationale for this limitation.
Response:
We sincerely thank the reviewer for this suggestion. We have separated the discussion of plant growth and abiotic stress and expanded the section on SiNPs’ effects on plant growth. In the newly added Section 2.2, we have also incorporated additional types of abiotic stress.
Comment 6:
- Relevance to animals: The manuscript repeatedly mentions studies in animals/mammals (e.g., lines 103–105, 136), despite being focused on plants. Clarify the relevance of these comparisons or remove them if not directly applicable.
Response 6:
In Lines 103–105, we intended to convey that nano‐silicon exerts an insecticidal effect via mechanical disruption, whereas in Line 136, we aimed to illustrate that, in the absence of a cell wall—as in animal cells—SiNPs can penetrate into the cytosol and trigger ROS bursts. Clarifying these points is essential for readers to understand both the toxicological mechanisms of SiNPs and the subsequent mechanistic discussions in this manuscript.
Comment 7:
- Missing details: Line 125 mentions “SiNPs, depending on particle size, surface area, concentration, pH, and plant species,” but subsequent discussion lacks detailed examples or mechanistic insights, especially regarding pH and plant 35 species. These should be elaborated upon.
Response 7:
We thank the reviewer for this suggestion. We re-examined the relevant reviews and conducted additional searches on the impact of pH on SiNP toxicity. Although soil pH has been reported to influence the toxicity of ZnO and TiO₂ (Integrated Environmental Assessment and Management, 3, 303–304), we identified no examples of pH-dependent SiNP toxicity. Accordingly, we have removed pH as an influencing factor in the revised manuscript.
Comment 8:
- Section 3: a) This section should be better systematized. b) Include a table listing influencing factors along with examples to enhance clarity.
Response 8:
We aimed here to clarify that SiNP size critically affects their uptake, distribution, and toxicity, leading us to infer that larger particles act primarily in the apoplast and, contrary to most reports, do not directly enhance antioxidant enzyme activity. Accordingly, and per the reviewer’s suggestion to improve clarity, we have retitled this section “The Size of SiNPs Profoundly Influences Distribution and Toxicity in Plants.” Because this topic is peripheral to the main focus of our review, we have not included a table in this section.
Comment 9:
- Section 5: This section discusses the role of ROS in plant growth and stress response, mainly focusing on H₂O₂. However, ROS also include ·O₂⁻, ·OH, O₃, and ¹O₂. Why are these species barely mentioned? Are they not involved, or is there insufficient data?
Response 9:
We thank the reviewer for this suggestion. In Lines 249–261, we have detailed the functions and characteristics of other ROS species and further emphasized the greater stability of H₂O₂, its role in systemic signaling, and the abundance of supporting studies.
Comment 10:
10.Section 5.1: As this section discusses SiNPs in salt stress, cold stress, and other abiotic stresses, a more specific and appropriate subheading is needed.
Response 10:
We have revised the subtitle to “5.1 Apoplastic H₂O₂ promotes plant growth and enhances tolerance to abiotic stresses by modulating intracellular redox homeostasis,” and organized the discussion into three sections addressing growth, abiotic stress, and biotic stress.
Comment 11:
- Section 5.2: The conclusion in lines 225–227 does not logically follow from the preceding discussion (lines 211–225). There seems to be missing information or a break in reasoning. Please revise for coherence.
Response 11:
We sincerely thank the reviewer for identifying this logical issue. We have removed, reorganized, and expanded the discussions in Sections 5.1 and 5.2 to enhance their readability and logical coherence.
Comment 12:
- Ensure all compounds are correctly formatted, e.g., H₂O₂ and Ca²⁺ (lines 228–232) should have proper sub/superscripts.
Response 12:
We apologize for these errors; they have now been corrected.
Comment 13:
- All abbreviations should be defined at first appearance, such as "ABC transporters" in line 243.
Response 13:
We have thoroughly addressed and revised this issue.
Comment 14:
- Section 6: The formatting of Section 6 should match previous sections. Avoid bullet points (“·”) and ensure 49 consistency in section style.
Response 14:
In this section, we have selected only a subset of questions for mechanistic speculation, with additional potential mechanisms discussed in Lines 397–400. We have removed all bullet points except those under “Unresolved Mechanistic Questions.” Furthermore, the entire section has been expanded in accordance with Reviewer 1’s comments.
Comment 15:
- The manuscript lacks summary visuals with only one figure. I would suggest the authors add relevant, informative figures which would improve clarity and readability:
(a) Include a table in Section 2 summarizing current research findings.
(b) Add a schematic diagram in Section 4 to illustrate proposed mechanisms.
Response 15:
We thank the reviewer for the suggestion. In Section 2, we have added a table illustrating representative examples of SiNP efficacy and underlying mechanisms. Additionally, in Section 4, we have included a schematic diagram depicting SiNP distribution and their effects on apoplastic functions.
Comment 16:
- Conclusion and Outlook: Please add a final section that summarizes the main conclusions of the review and 55 presents future perspectives. The outlook section should offer in-depth suggestions for future experimental directions, 56 which would increase the scientific value of the paper.
Response 16:
At the end of the manuscript, we have included the methods for identifying SiNP protein corona compositions and subsequent functional characterization, along with their significance. We also emphasize that understanding the initial sites of SiNP action is crucial for the development of novel agrochemicals and the identification of new genetic modification targets.
